# Study on the Ignition Mechanism of Inert Fuel Tank Subjected to High-Velocity Impact of Fragments

**DOI:** 10.3390/ma15093360

**Published:** 2022-05-07

**Authors:** Jian Liu, Fengjiang An, Cheng Wu, Longhui Zhang, Yanxi Zhang, Yipeng Li

**Affiliations:** State Key Laboratory of Explosion Science and Technology, Beijing Institute of Technology, Beijing 100081, China; 3120195176@bit.edu.cn (J.L.); chengwu@bit.edu.cn (C.W.); 3120205126@bit.edu.cn (L.Z.); 3120190270@bit.edu.cn (Y.Z.); 3220200133@bit.edu.cn (Y.L.)

**Keywords:** inert fuel tank, high-velocity fragments, gas concentration, ignition criterion, minimum ignition speeds

## Abstract

Nowadays, aircraft fuel tanks are protected by measures such as inerting, fire and explosion suppression, which significantly improve their ability to mitigate mechanical damage and prevent fire in the case of an accidental attack. In this study, an equivalent inert fuel tank with fire and explosion suppression was designed according to the vulnerabilities of a typical fighter. Then, a ballistic gun, a 37 mm gun and a two-stage light-gas gun were used to propel different fragments in tank damage experiments at different speeds (1400 m/s–2600 m/s). Experimental results show that the disassembly of a fuel tank is a prerequisite for igniting fuel. When the fragments hit the gas phase of the tank, the fuel tank was not disassembled and the fuel was not ignited. The calculation results show that the internal oxygen concentration was always lower than the limiting oxygen concentration (12%) before the fuel tank was disassembled. In addition, the minimum ignition speeds of inerted fragments with different masses as predicted by the ignition criterion when hitting the liquid fuel are consistent with the test results. This shows that increasing the mass of inert fragments will increase the minimum ignition speed and reduce the probability of ignition of the fuel. However, the implosion effect of the energetic fragments released about 3 times the chemical energy of its own kinetic energy, and the high-temperature and high-pressure products were very beneficial to the disintegration and ignition of the fuel tank compared to inert fragments.

## 1. Introduction

In recent years, energy and fuel have become research highlights. A large number of new technologies are applied to energy and fuel cells [1,2]. Importantly, extensive attention is paid to the safety of fuel in the industrial process and the reuse of green energy [3,4,5].

For aircraft, the fuel tank is an important component whose vulnerable areas account for more than 75% of the total volume [6]. Statistics show that the ignition and explosion of the fuel system are the main causes of aircraft damage and disassembly [7,8]. Therefore, adopting protective measures to prevent fuel tank ignition and explosion directly contributes to aircraft safety [9]. At present, the U.S. has actively taken inerting and explosion suppression measures for fuel tanks of commercial and military aircraft. The U.S. Air Force filled the fuel tanks of F-105, C-130 and F-4 with polyurethane foam in the shape of a net to prevent explosions. Later, the same method was used to protect the fuel tanks of A-7, A-10 and F-15 [10,11]. The U.S. military used an onboard inert gas-generating system (OBIGGS) to keep the oxygen concentration in the fuel tank below 9% (by volume) for C-17, F22, F-35 and A400 military transport aircraft to reduce the risk of fuel tank explosion due to projectile strikes [12].

A large number of studies have been carried out on the structural damage and ignition of unprotected fuel tanks using high-velocity inert and energetic fragments. Xie et al. [4] studied, through theoretical analyses and experiments, the damage and ignition of a full fuel tank by energetic fragments with a density of 7.8 g/cm^3^. Wang [13] studied the impact of reactive fragments and tungsten alloy fragments on a model fuel tank and the ignition of aviation kerosene, and found that the implosion and chemical energy release of reactive fragments are the main reasons for the serious damage of fuel tank structure and fuel combustion. Liu et al. [5] conducted an experiment on inert tungsten projectile impact on a fuel tank. The experimental results show that inert tungsten projectiles hardly result in effective structural damage to the fuel tank and effective ignition. Li et al. [14] introduced the ignition effect of inert fragments, tungsten and zirconium alloy fragments and reactive fragments on the fuel tank, and found that the comprehensive utilization of reactive fragment technology, velocity gradient technology and water hammer can improve the ignition probability of the target fuel tank. Through ballistic gun experiments and theoretical analyses and calculation, Wang et al. [15] studied the ignition effect of single energetic fragments on well-protected fuel tanks, and obtained the ignition mechanisms and ignition criterion. Moussa et al. [16] carried out a detailed theoretical analysis on the ignition effect of ordinary fragments on the fuel tank, the leakage speed of fuel and the evaporation speed of oil–gas in the fuel tank under different and intersecting conditions, and obtained the ignition probability of the fuel tank under different and intersecting conditions. Artero et al. [17] analyzed the influence of fuel levels in the tank on the damage of the fuel tank through numerical simulation and experiments.

However, the gas concentration, pressure change and oil–gas ignition conditions in inert fuel tanks are obviously different from those in unprotected fuel tanks subjected to accidental loadings [9,18,19,20,21,22,23,24]. The experiments carried out by the U.S. Army [18] showed that filling the fuel tank with polyurethane foam can significantly reduce the pressure inside the fuel tank after fragment and bullet impact and improve the possibility of survival of the fuel tank and aircraft. The OBIGGS is used to fill the fuel tank with inert gas to separate the fuel from the air, so as to form an explosion-proof medium in the fuel tank, which can also reduce the possibility of ignition of the fuel in the case of accidental load, and thus improve the safety of the fuel system [9]. Hill and Johnson [19] carried out an inerting experiment on full-scale fuel tanks and scaled fuel tanks, and believed that the experiment results for the full-scale fuel tank and the scaled fuel tank are consistent. Therefore, in order to reduce experimental costs, scaled fuel tanks can be used for the experiment; when the oxygen volume fraction is 9%, the fuel tank will not explode under any experiment conditions; an oxygen volume fraction of 12.5% is sufficient to protect the fuel tank under most ground fire conditions; in the case of fire caused by aircraft collision, an oxygen volume fraction of 18% only plays a limited role in protection. Anderson [20] established an experimental system, in which the fuel tank was inert with nitrogen and Halon 1301, and the protection of the fuel tank was tested with 23 mm HEIs (high-explosive incendiaries) as the ignition source. The results show that the overpressure generated by the reaction decreases sharply when the oxygen volume fraction is less than 14%, and disappears when the oxygen volume fraction is 9%; for the 750 L fuel tank used in the experiment, an obvious inerting effect can be produced by maintaining an oxygen volume fraction below 14%. Considering the actual tank size, shape, material and other factors, Anderson recommends that the limiting oxygen body fraction of aircraft fuel tank inerting be set to 10%. Tyson and Barnes [21] conducted a series of experiments at the U.S. Naval Weapons Center to evaluate the effect of oxygen volume fraction in oil-free space on combustion and explosion when the fuel tank of military aircraft is attacked by high-energy incendiary bombs. Shi et al. [22] proposed a new inerting analysis method to analyze the changes in the relationship of the internal oil–gas concentration of inert fuel tanks after bullet impact to judge the fire probability of the fuel tank. In addition, S Summer et al. [23] conducted an experiment to determine the reduction in oxygen concentration required to prevent fuel tank explosion, and obtained the critical oxygen concentration for the ignition of a fuel tank containing JP-8 under corresponding pressures at different altitudes of 0–38,000 feet above sea level. Ma Irimia [24] estimated the LOC (limiting oxygen concentration) of fuel, air and an inert mixture at a high temperature according to the adiabatic flame temperature, and found that adding inert components to the fuel–air mixture determines the increase in LEL (lower explosion limited) and the decrease in UEL (upper explosion limited), until these values are finally combined at the inert point.

On a real battlefield, the relative speed of the fuel tank and fragments may reach up to 2–3 km per second. It is very necessary to study the internal oil–gas concentration, pressure and temperature change and ignition of inert fuel tanks subjected to high-velocity fragments. However, at present, the research on high-velocity (especially above 2000 m/s) fragment damage of inert fighter fuel tanks is insufficient, and no clear and effective experiment conclusions or criteria have been obtained, which is unfavorable for the structural design of aircraft fuel tank and the optimization of missile warheads. According to the vulnerability characteristics of typical fighters, an equivalent model with an inerting and protected fuel tank was designed in this study. Different inert fragments and energetic fragments were driven by a ballistic gun, a 37 mm gun and a two-stage light-gas gun, and the results of damage to the inert fuel tank under different speeds (1400 m/s–2600 m/s) were obtained. When the fragments hit the gas phase of the tank, the fuel tank was not disassembled and the fuel was not ignited. The calculation results show that the internal oxygen concentration was always lower than the LOC (12%) before the fuel tank was disassembled. Finally, the minimum ignition speeds of inert fragments with different masses were predicted by using the ignition criterion. Increasing the mass to improve the fragment kinetic energy so as to aggravate the mechanical damage of the fuel tank also increased the minimum ignition speeds of the fragments, which is not conducive to igniting the fuel. However, the implosion effect of energetic fragments greatly increased the damage and ignition effects of the fuel tank. Theoretical calculation results accurately explain and verify the experiment results, which provide support for the warhead design of air defense missiles and the damage evaluation of fuel tank targets.

## 2. Experimental Setup

### 2.1. Equivalent Inert Fuel Tank

Previous research shows that, after the vulnerability analysis experiment and several improvements, the only part of F-22 still unable to deal with threat targets is the pilot cockpit. In the cockpit, the F-1 fuel tank is behind the pilot’s seat. The pilot’s safety and seat ejection and other functions are seriously threatened when the F-1 fuel tank is ignited or detonated. It is an important component in the low-vulnerability design of F-22, for which live-fire analyses and experiments [25,26,27] have been carried out. Therefore, the wing fuel tank A-2 of F-22 and the F-1 fuel tank near the pilot seat were preliminarily determined as the key research objects, and an equivalent fuel tank model with a dry cabin structure, as shown in Figure 1, was established. In the middle was the oil bunker, which was filled with polyurethane for flame and explosion suppression; the outer layer was a dry chamber structure filled with polyurethane foam. The fuel tank structure was designed according to the ultimate penetration speed principle. The fuel tank material was LY-12 aluminum alloy with a shell thickness of 3 mm, which was made by welding. The size of the space inside the fuel tank into which oil was poured was 400 mm × 400 mm × 400 mm. It was expected that the fuel load of the aircraft entering the theater is usually 50–60%. Therefore, the fuel tank load in the experiment was 60%. As shown in Table 1, the physicochemical properties of RP-3 kerosene are similar to those of JP-8. So, the fuel in the tests was RP-3 aviation kerosene.

Based on fighter fuel tanks, inerting measures were conducted in the experiments. First, mesh polyurethane foam was stuffed into both dry and oil storage tanks. Second, in order to simulate the inerting environment with less oxygen, after filling the oil bunker with kerosene, the inside of the fuel tank was washed with nitrogen for no less than 2 min until the oxygen concentration at the oil filling hole was 9% (±0.1%), as measured by the ITG I-01 oxygen concentration sensor.

### 2.2. Fragments and Acceleration Devices

The inert fragments and energetic fragments used in the tests are shown in Figure 2. The inert fragments were made of steel with masses of 4, 7, 10, 23 and 35 g. Energetic fragments can be divided into 3.3, 5, 7 and 9 g.

As shown in Figure 3, a ballistic gun, a 37 mm gun and a two-stage light-gas gun were used to shoot various fragments. The ballistic gun was used when the impact velocity of fragments was meant to reach 1400–1500 m/s. The 37 mm gun was used to accelerate fragments at the velocity of 1700 m/s–1900 m/s. Fragments reached a speed over 2000 m/s when fired from the two-stage light-gas gun.

### 2.3. Experimental Principles

The test principle is shown in Figure 4, which is composed of the fragment acceleration device, fragments, a sabot recycler, a velocity probe and a fuel tank. The sabot recycler was used to recycle the separated sabots after acceleration. The impact speeds of fragments were measured by the velocity probe in front of the tank. The damage to the fuel tank caused by fragments was recorded by a high-speed camera so the damage process and mechanism could be explored.

## 3. Experimental Results

The damage experiments using inert fragments and energetic fragments at different speeds were carried out for the equivalent fuel tank of a typical fighter F-22, and 15 effective experiment results were obtained, which are shown in Table 2. Among them, only the energetic fragment in test No.12 hit the gas phase space. Additionally, the other fragments all hit the liquid fuel in the tank.

It can be seen from Table 2 that only one of the nine steel fragments in the damage experiments for the inert fuel tank ignited the fuel, and the other eight did not; among the five energetic fragments, three ignited the fuel, and the other three did not.

When the 4 g steel fragments hit the inert fuel tank at the speeds of 1461 m/s and 1497 m/s (the kinetic energy was 4269 J and 4482 J, respectively), they only produced perforation and slight deformation and did not ignite the fuel. As shown in Figure 5 and Figure 6b, when the 7 g steel fragments impacted the fuel tank at the speed of 1868 m/s (the kinetic energy was 12,212 J), the fragment impacted the fuel tank surface and produced a large amount of firelight, and then entered the fuel tank, and the penetrated side was deformed under the influence of water hammer. Then, the liquid fuel formed a jet at the hole of the penetrated side, and atomized in the air together with the fuel splashed on the crack, but was not ignited. When the 7 g steel fragment hit the fuel tank at a high speed of 2611 m/s (the kinetic energy was 23,860 J), the deformation of the fuel tank was more serious, and a large area around the bottom weld cracked and fell off, but the oil–gas was not mixed with the air in time and could not be ignited. After the 23 g steel fragments hit the fuel tank at the speed of 1885 m/s (the kinetic energy is 40,862 J), the penetrated side of the fuel tank bulged and deformed seriously, the upper end cover was lifted and flew up, and the oil–gas was fully mixed with air, but the fuel could not be ignited. As shown in Figure 7 and Figure 8, with the continuous increase in the kinetic energy of inert fragments, the 35 g steel fragment hit the tank shell at the speed of 2298 m/s (the kinetic energy is 92,414 J). Under water hammer generated by the high-speed fragments, the tank was seriously deformed and disassembled, and a large amount of liquid fuel was splashed, gasified and quickly mixed with air to reach the combustion concentration range. The oil–gas was ignited rapidly due to the high temperature of the fragment, while the liquid fuel was also ignited, and the fuel tank was completely ignited and detonated. It can be seen from the results that the disassembly of the fuel tank was a necessary condition for igniting fuel, and kinetic energy was an important factor for the damage and ignition of the fuel tank.

There were few reactive materials in the energetic fragments with small masses, so the energy released after impact was relatively low. As shown in Figure 9, after the energetic fragments of 3.3 g and 5 g hit the fuel tank at the speeds of 1812 m/s and 1783 m/s, respectively, although the energetic fragments were effectively excited, the internal pressure generated by water hammer and the implosion effect were not enough to disassemble the fuel tank. The deformation of the fuel tank only appeared on the penetrated side. Cracks were generated at the weld, and the fuel was not ignited.

When 7 g energetic fragments hit the gas phase space of the fuel tank at the speed of 1857 m/s (the kinetic energy is 12,070 J) in test No.12, the fuel tank was not obviously deformed. There was a circular hole on the penetrating face and back face. When hitting the gas phase, the water hammer effect was not generated. So, the fuel tank was not severely deformed, and the fuel ignited.

However, when the 7 g and 9 g energetic fragments hit the liquid fuel in the tank in tests No.13, 14 and 15, they ignited the fuel reliably. As shown in Figure 10, after the 9 g energetic fragments hit the fuel tank surface, bright firelight was produced, and then the penetration surface bulged outward under the great pressure, resulting in cracking and deformation of the weld. Subsequently, the upper end cap of the fuel tank was also lifted under the combined influence of the water hammer and implosion effects, and the liquid fuel evaporated into oil–gas, which quickly and fully mixed with the air. The reactive material inside the fragment excited the chemical reaction to produce high-temperature and high-pressure products, which quickly ignited the fuel.

In summary, the fuel tank was inert and oxygen-poor. For the two types of fragments, the disassembly of the fuel tank was a necessary condition for igniting fuel. The degree of damage to the fuel tank caused by inert fragments mainly depended on the kinetic energy of fragments, while that caused by energetic fragments depended more on the amount of internal reactive material. As in test No. 12, even though the energetic fragments hit the gas phase space of the fuel tank, the fuel tank was not severely deformed or ignited. On the contrary, the pressure produced by the combination of water hammer and the implosion effect of energetic fragments in tests No. 13 and 14 was high and thus easily disintegrated the fuel tank and made the mixture of oil–gas and air reach the ignition concentration range. At the same time, high-temperature and high-pressure reaction products ignited oil–gas more effectively. More importantly, about 35,000 J of kinetic energy was required to disintegrate the inert fuel tank. In addition, by comparing the experiment results of tests No. 7, No. 8, No. 11 and No. 13, it was found that the energetic fragments released about 3 times the chemical energy of its own kinetic energy.

## 4. Ignition Mechanisms

### 4.1. Gas Concentration Change

The ignition of fuel requires meeting three conditions: oil–gas within the combustible concentration range, appropriate oxygen levels and an ignition source with sufficient energy. Figure 11 shows the flammability of RP-3 fuel vapor.

In Figure 11, U and L are the upper and lower limits of the combustion concentration of RP-3 fuel vapor in pure oxygen, respectively. The straight-line CA is the air line, and the corresponding points U and L are the upper and lower limits of combustion concentration in the air. The straight-line Uu and the straight-line Ll were extended to obtain the intersection E. The red-shaded part in Figure 11 is the combustible area. The oxygen concentration corresponding to point E is the limiting oxygen concentration (LOC). When the oxygen concentration is lower than the LOC, no matter whether the oil–gas concentration is within the combustible concentration range, the fuel cannot be ignited. FAA has studied the LOC that does not support combustion in the fuel tank, and found that the oxygen concentration does not exceed 12% from sea level to 3048 m (10,000 feet), linearly increasing from 12% to 14.5% from 3048 m (10,000 feet) to 12,192 m (40,000 feet), and can be linearly extrapolated above 12,192 m (40,000 feet) [28].

Before the experiment, the inside of the fuel tank was inert and was in a state of rich nitrogen and poor oxygen. In the process of air washing, the fuel tank was always under normal temperature and pressure conditions, which can be regarded as ideal gas. Assuming that the volatilization rate of oil–gas is equal to the exhaust rate, according to the ideal gas equation of state, the following is obtained:(1)$$\frac{V_{f}}{V_{u}}=\frac{n_{f}}{n_{u}}=\frac{P_{f}}{P_{u}}$$
where $V_{f}$, $n_{f}$ and $P_{f}$ are the volume of oil–gas, the amount of material and the saturated vapor pressure in the tank, respectively. $V_{u}$, $n_{u}$ and $P_{u}$ are the volume of gas, and the amount and pressure of gas substances, respectively. Therefore, the volume fraction of oil–gas in the mixed gas in the tank was equal to:(2)$$\frac{V_{f}}{V_{u}}=\frac{P_{f}}{P_{u}}=\frac{4.25\;\mathrm{kPa}}{103.25\;\mathrm{kPa}}=4.12\%$$

Therefore, the initial volume fractions of oxygen, oil–gas and nitrogen in the inert fuel tank were 9%, 4.12% and 86.88%, respectively.

There may be many paths for fragments to enter the fuel tank. For the convenience of analysis, it is assumed that the fragments followed the two paths shown in Figure 12 in this study. According to the test results, even though energetic fragments entered the fuel tank according to shot line 1 (meaning that fragments entered the gas phase space), the fuel tank was not severely deformed or ignited. Therefore, it mainly discusses changes in the concentration of oil–gas and oxygen and the ignition of fuel after the fragments entered the liquid phase space.

When the fragment entered the fuel tank following shot line 2 in Figure 12, the fragment brought a large amount of air into the tank and formed a hole in the liquid fuel. The volume of air is related to the cross-sectional area of the fragment, the moving speed of the fragment and the moving distance in the tank. After the fragment entered the liquid fuel and moved a certain distance, the liquid fuel blocked the perforation of the fragment, and air could not enter the liquid fuel. This distance was about 3 times the diameter of the fragment [29].
(3)$$S_{L}=3D$$
(4)$$V_{air}=\frac{1}{4}\pi D^{2}V_{r}S_{L}=\frac{3}{4}\pi D^{3}V_{r}$$
where D is the diameter of the fragment, $S_{L}$ is the moving distance of the fragment when the air stopped entering the fuel tank, $V_{air}$ is the volume of the air entering the fuel tank, and $V_{r}$ is the residual velocity of the fragment passing through the fuel tank surface, which can be obtained according to Equation (5) [30]:(5)$$V_{r}=\frac{\sqrt{V^{2}-V_{50}{}^{2}}}{1+\frac{\rho_{b}A_{p}b}{m\cos\theta }}$$
(6)$$V_{50}=\sqrt{\frac{\pi DSb^{2}}{m\cos^{2}\theta }}$$
where V is the impact velocity of the fragment, $V_{50}$ is the penetration limit velocity, $\rho_{b}$ is the density of the fuel tank shell, $A_{p}$ is the facing area of the fragment, b is the thickness of the fuel tank shell, m is the fragment mass, θ is the incident inclination angle, and S is the shear resistance.

When fragments entered the liquid fuel, they were subject to the action of liquid resistance to slow them down. According to Newton’s second law:(7)$$m\frac{dV}{dt}=-F_{D}=-A_{P}C_{D}\rho_{l}\frac{V^{2}}{2}$$
where $F_{D}$ is the resistance, $\rho_{l}$ is the density of fuel and $C_{D}$ is the resistance coefficient. Assuming that the resistance coefficient in the differential Equation (7) is constant, the velocity expression is obtained:(8)$$V(t)=\frac{V_{r}}{1+\frac{DLC_{D}\rho_{l}}{2m}V_{r}t}$$
where L is the length of the fragment. The high-temperature fragment transfers heat to liquid fuel:(9)$$q^{\prime \prime }=-\frac{mc}{\pi DL}\frac{dT}{dt}$$
where $q^{\prime \prime }$ is the energy flux, c is the specific heat of the fragment, and c=460 J/kg·K. Therefore, the mass of oil–gas generated by the transferred heat is:(10)$$m^{\prime \prime }=\frac{q^{\prime \prime }}{c_{f}(T_{b}-T_{0})+h_{q}}$$
where $c_{f}$ is the specific heat of the fuel, and $T_{b}$, $T_{0}$ and $h_{q}$ are the boiling point, initial temperature and evaporation heat of the fuel, respectively.

Due to the high thermal conductivity of fragments, the temperature gradient inside fragments can be ignored. When passing through the fuel, the fragment temperature T is a function of time, which can be deduced as [4]:(11)$$T=T_{0}+(T_{S}-T_{0})e^{\left(-\frac{6ht}{D\rho c}\right)}$$
where $T_{0}$ is the ambient temperature, and $T_{0}=298\;K$ here; $T_{S}$ is the temperature when the fragment passes through the fuel tank shell, and h is the heat transfer coefficient [16].
(12)h=kD[0.3+0.62ReD12Pr13[1+(0.4Pr)23]14[1+(ReD28200)58]45]
where k is the conductivity coefficient of fuel, and $P_{r}$ is the Prandtl constant. For RP-3, k=0.145 W/(m·K), $P_{r}=17$ [16], and ReD is the Reynolds number:(13)ReD=DV(t)ν
where ν is the kinematic viscosity coefficient of fuel, and for RP-3 fuel, the value of ν is 2.37 mm^2^/s [4]. When fragments penetrate the fuel tank shell, there will be speed loss. Ignoring the heat dissipation, the lost kinetic energy will be converted into the internal energy Q of fragments [31]:(14)$$Q=\beta W=\beta \frac{1}{2}m(V^{2}-V_{r}{}^{2})$$
where β represents the plastic products that are transformed into heat. Mason [32] experimented with the temperature rise during plastic deformation and found that the strain rate of aluminum 2024-T3 alloy was in the range of 0.5–0.9 at 3000 s^−1^. For simplicity’s sake, the value used in this paper is 0.5. In order to maximize the impact on the concentration of oil–gas components in the tank, it is assumed that all heat is transferred to fragments:(15)$$(T_{S}-T_{0})mc=Q$$

Fragments change the proportion of oil–gas, oxygen and nitrogen in the fuel tank. The amounts of substance of various gases in the new mixed gas are as follows:(16)$$\begin{array}{l} n_{o_{2}}=\frac{V_{u}\times 9\%}{0.0224}+\frac{V_{air}\times 21\%}{0.0224}\\ n_{N_{2}}=\frac{V_{u}\times 86.88\%}{0.0224}+\frac{V_{air}\times 79\%}{0.0224}\\ n_{fuel}=\frac{V_{u}\times 4.12\%}{0.0224}+\frac{m_{vap}}{M} \end{array}$$
where $m_{vap}$ is the mass of newly converted oil–gas, and M is the molar mass of RP-3, whose value is 148.
(17)$$m_{vap}=\int_{0}^{t}-m^{\prime \prime }dt=\int_{0}^{t}\frac{-q^{\prime \prime }}{c_{f}(T_{b}-T_{0})+hq}dt$$

Here, *t* is the flight time of the fragment. At the same time, for an ideal mixture, the mole fraction of steam is equal to the volume fraction. The oil–gas concentration and oxygen concentration (volume fraction) shown in Figure 13 and Figure 14 can be calculated from Equation (16).

The curves in the non-shaded parts in Figure 13 and Figure 14 represent the variations in oil–gas and oxygen concentrations in the fuel tank, respectively (the shaded part indicates that the fuel tank has been disassembled and is no longer suitable for the variation in gas concentration in this model). In tests No. 1–7, the oxygen concentration inside the tank was lower than the LOC (12%), so the fuel could not be ignited. This is consistent with the test results. In addition, it can be seen from Figure 13 that under the high-velocity impact of inert fragments, the oil–gas concentration in the fuel tank increased with the increase in the mass and speed. On the contrary, the oxygen concentration in the tank increased first and then decreased slightly with the increase in fragment speed. This is because when the speed is low, more air is brought into the tank by fragments. With the increase in speed, the heat generated by the impact converts into more oil–gas, so the oxygen concentration decreases relatively. Then, the increase in oxygen concentration caused by inert fragments with a high mass entering the fuel tank is higher than that caused by low-mass fragments. However, in all cases, the oxygen concentration in the tank was always lower than the LOC (12%), and thus in the state of oxygen deficiency and unable to ignite. Therefore, it is verified that the disassembly of the inert fuel tank is a necessary condition for igniting fuel.

### 4.2. Ignition Criteria

Johnson [16] (1988) studied the effect of hot parts on fuel ignition. The ignition of kerosene not only requires a high enough temperature, but also for that temperature to be maintained for the duration of the ignition delay time. By combining the ignition criterion of fuel–air mixture with the Arrhenius formula, the ignition delay time can be described as:(18)$$t_{c}=f\cdot \frac{\exp(E_{a}/RT)}{p^{n}}$$
where *f* is the pre-exponential factor, *E_a_* is the activation energy (the activation energy of hydrocarbons is almost the same), *p* is the pressure, *R* is the universal gas constant, *T* is the absolute temperature, and *n* is the reaction state. For Jet A fuel, the ignition delay time is the same as that of JP-8 fuel. The lower the temperature of the fuel, the lower the pressure, and the longer the combustion hysteresis time. For RP-3, *f* is 1.68 × 10^−8^ ms/atm^2^, *E_a_* is 158.2 kJ/mol and *n* is 2 [16].

According to Equations (11) and (18), the ignition delay time of inert fragments with different masses and different speeds can be calculated, and then the flight time of fragments in the tank can be obtained by integrating Equation (8). Figure 15 was obtained by comparing the flight time with the ignition delay.

It can be seen in Figure 15 that for fragments with the same mass, the speed had a greater impact on the ignition delay time than the flight time. This is because the higher the speed, the higher the fragment temperature after impact, which will greatly reduce the ignition delay time. In addition, the ignition delay time of fragments with smaller masses was shorter at the same speed. This is because the penetration limit velocity ($V_{50}$) of small-mass fragments is greater, which means that the fragment loses more kinetic energy, resulting in a higher fragment temperature. Points A, B, C, D and E in Figure 15 are the minimum speeds and ignition delay times required for 4 g, 7 g, 10 g, 23 g and 35 g inert fragments to ignite fuel. Through the analysis of Figure 13, Figure 14 and Figure 15, it can be said that for 4 g, 7 g and 10 g fragments in tests, the kinetic energy was too low to disintegrate the fuel tank, and the fuel in the tank was in a state that could not be ignited. The minimum speeds of fuel tank disassembly by fragments of these three qualities were significantly higher than that of ignition. Therefore, when the impact velocity of these three fragments was increased to disintegrate the fuel tank, the fuel was ignited at the same time; in test No. 8, the 23 g inert fragment (1885 m/s) disintegrated the fuel tank, but it moved slower than the minimum ignition speed (2081 m/s) and could not ignite the fuel; on the contrary, in test No. 9, the 35 g inert fragment (2298 m/s), whose speed was greater than its minimum ignition speed (2246 m/s), met the ignition criterion and destroyed the fuel tank, and the fuel was ignited. It is consistent with the results.

### 4.3. Comparisons of Fuel Ignition Mechanism between Inert Fragments and Energetic Fragments

Through experiments and analyses, it was found that the deformation and fracture of fuel tanks are caused by water hammer for inert fragments. On the one hand, increasing the kinetic energy of fragments can significantly increase the degree of mechanical damage of the tank. On the other hand, increasing the fragment mass will correspondingly increase the ignition delay time. So, inert fragments with appropriate speeds and qualities should be selected.

However, fuel tanks more easily deform and disintegrate when hit by energetic fragments under the combined action of implosion and water hammer. At the same time, the temperature of the reaction of reactive materials can reach up to 3000 K, which greatly reduces the ignition delay time and makes it easier to ignite fuel. Increasing the mass of energetic fragments can increase the pressure inside the tank, which contributes to the deformation and disassembly of the fuel tank. Increasing the speed of fragments can increase the energy release efficiency of reactive materials. Therefore, compared with inert fragments, energetic fragments with lower speeds and smaller masses make it easy to ignite an inert fuel tank.

## 5. Conclusions

In this paper, the ignition mechanism of an inert fuel tank was studied experimentally and theoretically. Firstly, through the analysis of vulnerability characteristics of fuel tanks, an equivalent inert fuel tank was designed in this study. Then, the tank damage experiments, in which fuels tanks were subjected to inert fragments and energetic fragments at different speeds, were carried out using three kinds of acceleration devices, with which damage and ignition results of inert fuel tanks were obtained. Finally, through theoretical analysis, the variation in gas concentration in the fuel tank subjected to different inert fragments was obtained, and the minimum ignition speeds of fragments with different masses were predicted according to the ignition criterion. The main conclusions of this study are as follows:

(1) The disassembly of the fuel tank is a necessary condition for igniting fuel. The calculation results show that the oxygen concentration in the fuel tank is lower than the LOC (12%) before the disassembly. Therefore, neither inert fragments nor energetic fragments can disintegrate the fuel tank and ignite the fuel when the fragments hit the gas phase space of the tank.

(2) It can be seen from the test results that the kinetic energy from inert fragments is the dominant actor in causing mechanical damage to the fuel tank, and about 35,000 J of kinetic energy is required to disintegrate the inert fuel tank when hitting the liquid phase space of the tank. Due to the implosion effect, the energetic fragment can release about 3 times the chemical energy of its own kinetic energy, which significantly increases the damage to the fuel tank.

(3) According to the ignition criteria, the minimum ignition speeds of 4 g, 7 g, 10 g, 23 g and 35 g inert fragments are 1479 m/s, 1570 m/s, 1752 m/s, 2081 m/s and 2246 m/s, which is consistent with the test results. It shows that an increase in the mass of inert fragments will increase the minimum ignition speed, but small-mass fragments are detrimental to the disassembly of the fuel tank. Therefore, inert fragments with appropriate qualities and speeds should be selected to ignite fuel.

However, the high-pressure and high-temperature products produced by the implosion effect of the energetic fragments greatly enhanced the disintegration and ignition of the fuel tank compared to inert fragments. Therefore, when the energetic fragments hit the fuel phase and disintegrated it, the fuel was ignited stably.

## Figures and Tables

**Figure 1 materials-15-03360-f001:**
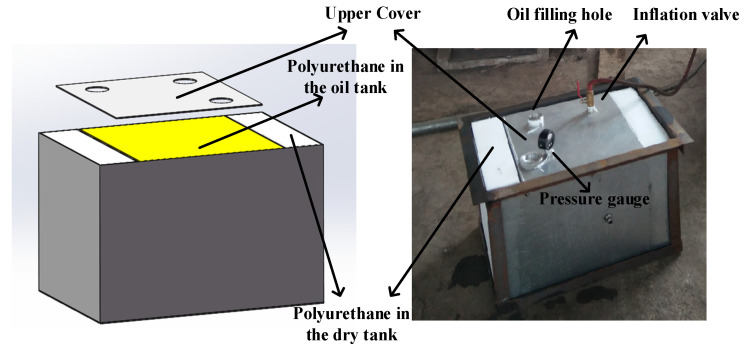
Model and real equivalent fuel tank of F-22 fighter aircraft.

**Figure 2 materials-15-03360-f002:**
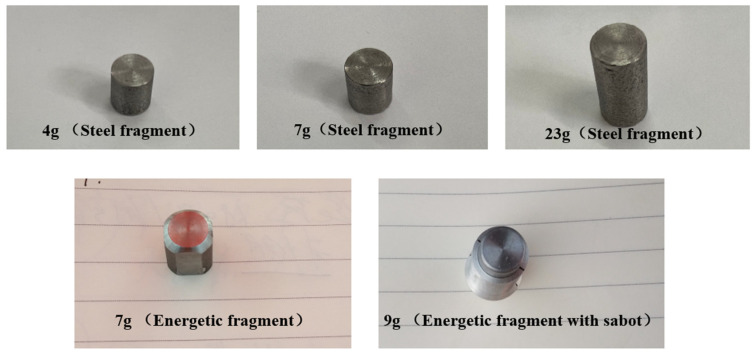
Inert steel fragments and energetic fragments.

**Figure 3 materials-15-03360-f003:**
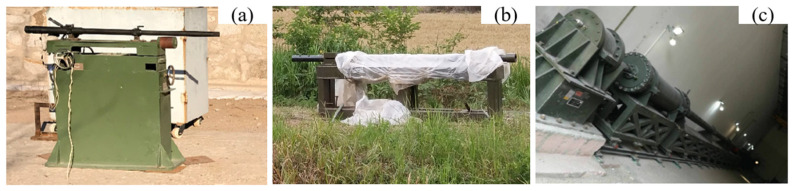
Three fragment acceleration devices. (**a**) A ballistic gun; (**b**) a 37 mm gun; and (**c**) a two-stage light-gas gun.

**Figure 4 materials-15-03360-f004:**
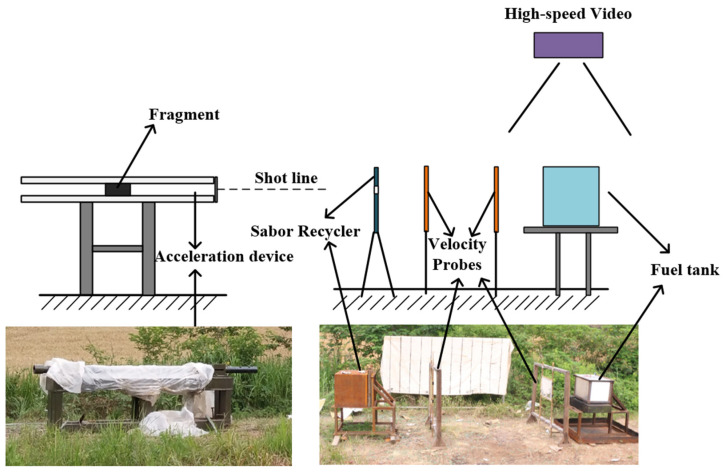
Schematic and photographs of experimental setup.

**Figure 5 materials-15-03360-f005:**
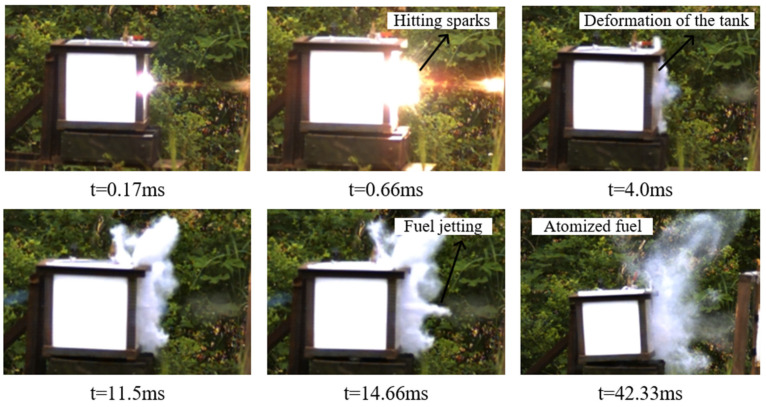
High-speed video frames of the fuel tank damaged by the 7 g steel fragments (v = 1868 m/s).

**Figure 6 materials-15-03360-f006:**
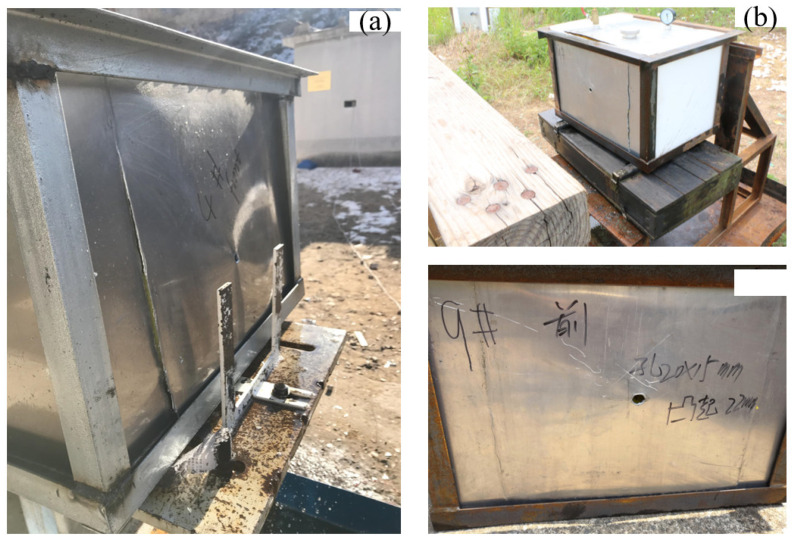
Damage to the fuel tank caused by steel fragments at medium and low speeds. (**a**) Damage to the fuel tank caused by the 10 g steel fragments at 1454 m/s; and (**b**) damage to the fuel tank caused by the 7 g steel fragments at 1868 m/s.

**Figure 7 materials-15-03360-f007:**
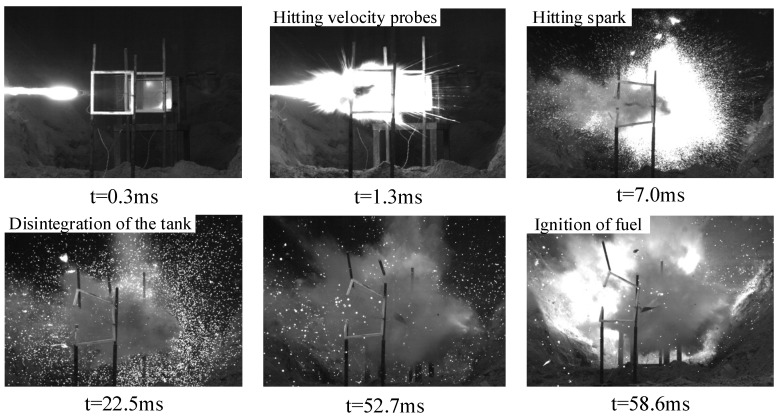
High-speed video frames of the fuel tank damaged by the 35 g steel fragments (v = 2298 m/s).

**Figure 8 materials-15-03360-f008:**
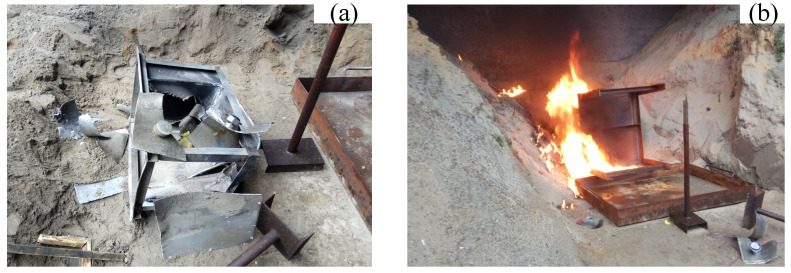
Damage and destruction of the fuel tank caused by the 35 g steel fragments (v = 2298 m/s). (**a**) Serious deformation and disassembly of the fuel tank; and (**b**) ignition and detonation of fuel and the fuel tank.

**Figure 9 materials-15-03360-f009:**
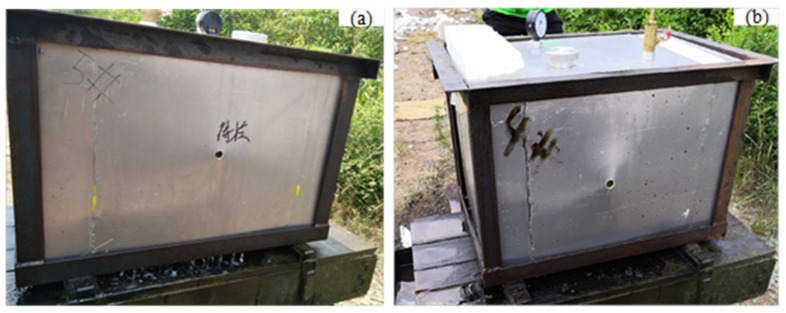
Damage to the fuel tank caused by small-mass energetic fragments. (**a**) Damage to the fuel tank caused by the 3.3 g energetic fragments; and (**b**) damage to the fuel tank caused by the 5 g energetic fragments.

**Figure 10 materials-15-03360-f010:**
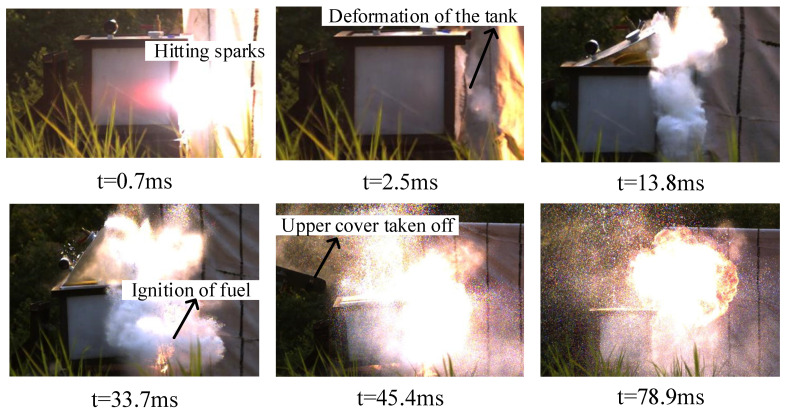
Destruction of the fuel tank by the 9 g energetic fragment (v = 1781 m/s).

**Figure 11 materials-15-03360-f011:**
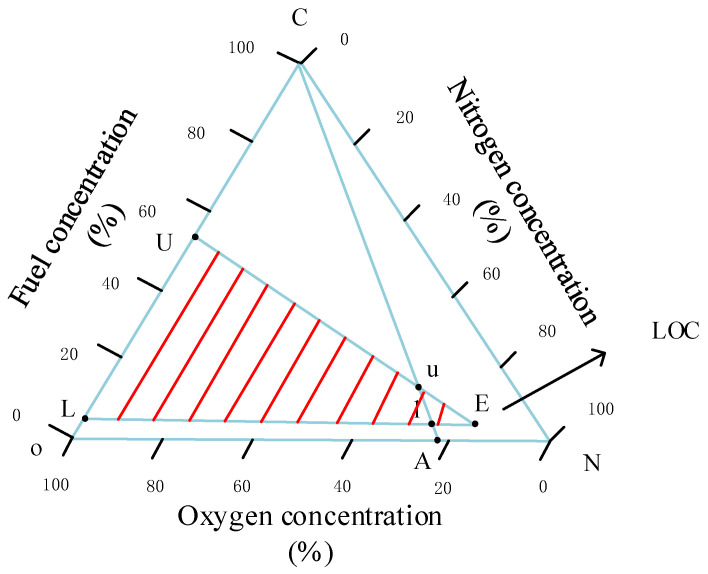
RP-3 fuel flammability diagram.

**Figure 12 materials-15-03360-f012:**
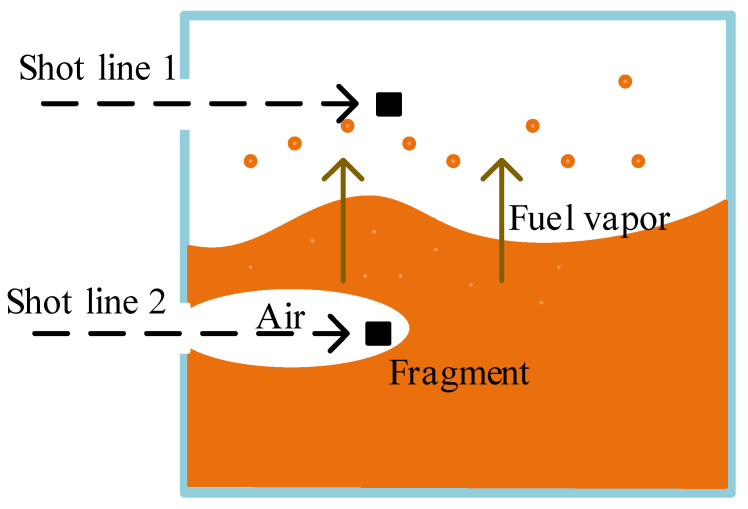
Gas changes after the fragment entered the fuel tank.

**Figure 13 materials-15-03360-f013:**
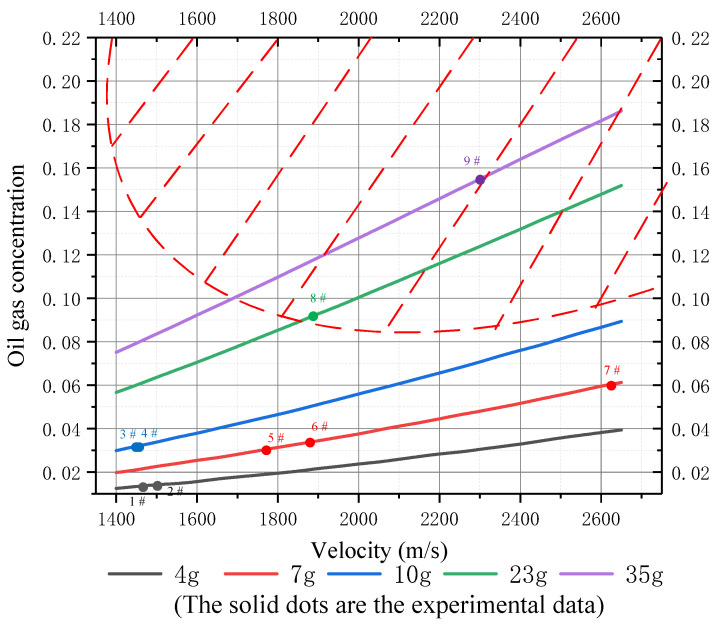
Variation in oil–gas concentration (volume fraction) in the fuel tank with velocity subjected to inert fragments with different masses.

**Figure 14 materials-15-03360-f014:**
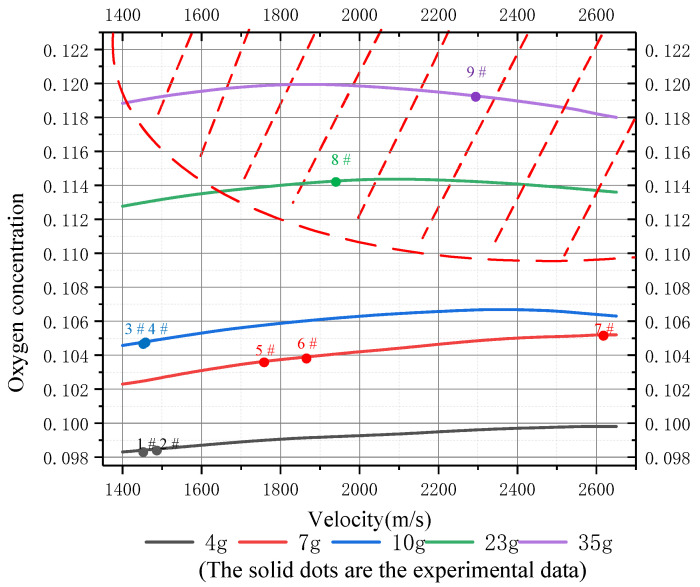
Variation in oxygen concentration (volume fraction) in the fuel tank with velocity subjected to inert fragments with different masses.

**Figure 15 materials-15-03360-f015:**
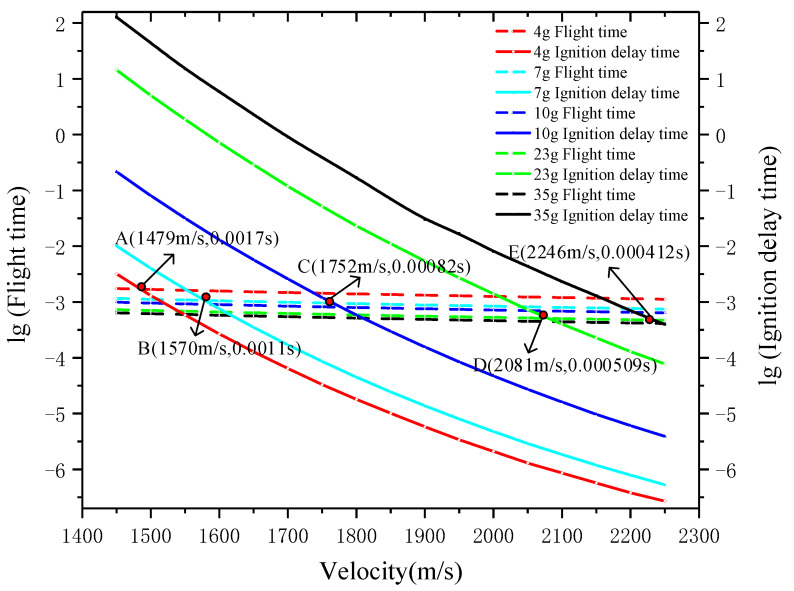
Comparison of flight time and ignition delay time with different velocities and masses of Inert Fragments.

**Table 1 materials-15-03360-t001:** Physicochemical properties of RP-3 and JP-8 [4,16,22].

Physicochemical Properties	RP-3	JP-8
Density (g/cm^3^)	0.78	0.78
Saturated vapor pressure at 25 °C (kPa)	4.25	4.21
Flash point (K)	311	311
Freezing point (K)	226	226
Viscosity at 20 °C (mm^2^/s)	1.25	1.25
Heat of evaporation (KJ/kg)	345	291
Net heat of combustion (MJ/kg)	42.8	42.8
Combustion concentration limit	0.83–5.68%	0.6–4.7%
Critical oxygen concentration	12%	12%

**Table 2 materials-15-03360-t002:** Experimental results.

Fragments	Number	Masses (g)	Speeds (m/s)	Kinetic Energy (J)	Results
Inert fragments (steel)	1	4	1461	4269	The fuel tank was perforated, the perforation was slightly deformed, and the fuel was not ignited.
	2	4	1497	4482	
	3	10	1455	10,585	The fuel tank was perforated and obviously deformed, the weld was cracked greatly, and the fuel was not ignited.
	4	10	1454	10,570	
	5	7	1762	10,866	
	6	7	1868	12,212	
	7	7	2611	23,860	The fuel tank was perforated, bulging and deformed, the bottom weld was serious cracked and fell off, and the fuel was not ignited.
	8	23	1885	40,862	The penetrated side of the fuel tank was bulging and deformed, cracking the weld, the upper cover was lifted, and the fuel was not ignited.
	9	35	2298	92,414	The fuel tank was completely destroyed and disassembled, the oil–gas was quickly mixed with air, and the fuel was ignited.
Energetic fragments	10	3.3	1812	5418	The penetrated side of the fuel tank was broken, and the weld was cracked due to bulging; the tank was relatively intact, the fuel was not ignited.
	11	5	1783	7948	
	12	7	1857	12,070	The energetic fragment hit the gas phase space of the tank. There was a circular hole on the penetrating surface. The fuel tank was not obviously deformed. Additionally, there was also a hole in the back.
	13	7	1762	10,866	The upper cover was lifted directly, the surrounding area of the penetrated side cracked and fell off, there were large cracks in the bottom weld, and the fuel was ignited.
	14	7	1875	12,304	
	15	9	1781	14,273	

## Data Availability

The data used to support the findings of this study are included within the article.

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
