# Peer review of "Study on the Ignition Mechanism of Inert Fuel Tank Subjected to High-Velocity Impact of Fragments"

_materials, 2022, doi:10.3390/ma15093360_

Round 1
Reviewer 1 Report
-Figures 2 to 5 should increase in size and quality.
-Nomenclature must be added.
-Check the text for font size, notation, etc.
-Systematically rewrite the introduction.
-I recommend adding a general paragraph at the beginning of the introduction about new technologies. New technologies including related to your work and new technologies in fuel and energy supply with fuel cells:
*Experimental investigation on enhanced damage to fuel tanks by reactive projectiles impact
*Modeling of a high temperature heat exchanger to supply hydrogen required by fuel cells through reforming process
*Enhanced ignition behavior of reactive material projectiles impacting fuel-filled tank
*CFD simulation of time-dependent oxygen production in a manifold electrolyzer using a two-phase model
*Energy, exergy, environmental and economic analyzes (4E) and multi-objective optimization of a PEM fuel cell equipped with coolant channels
Author Response
Response to Reviewer 1 Comments
Point 1: Figures 2 to 5 should increase in size and quality.
Response 1: We are so sorry for the quality and size issues in Figures 2 to 5. And we have tried our best to improve them in the revised manuscript. However, pictures in Figure 5 from high-speed video will still be somewhat blurry due to the equipment.
Point 2: Nomenclature must be added.
Response 2: We are sorry for your confusion about some technical terms. We have named and explained some mathematical symbols in Appendix at the end of the paper.
Point 3: Check the text for font size, notation, etc.
Response 3: We are very sorry for our negligence of errors of the text for size and notation. And We have carefully corrected the text and marked them in red.
Point 4: Systematically rewrite the introduction.
Response 4: I have rewritten the part of introduction of the paper based on the comments of the reviewers.
Point 5: I recommend adding a general paragraph at the beginning of the introduction about new technologies. New technologies including related to your work and new technologies in fuel and energy supply with fuel cells:
*Experimental investigation on enhanced damage to fuel tanks by reactive projectiles impact
*Modeling of a high temperature heat exchanger to supply hydrogen required by fuel cells through reforming process
*Enhanced ignition behavior of reactive material projectiles impacting fuel-filled tank
*CFD simulation of time-dependent oxygen production in a manifold electrolyzer using a two-phase model
*Energy, exergy, environmental and economic analyzes (4E) and multi-objective optimization of a PEM fuel cell equipped with coolant channels
Response 5: This is a very insightful comment. And I have added an introduction to new technologies in fuel and energy supply with fuel cells as the reviewer’s comments.
Special thanks to you for your good comments.

Reviewer 2 Report
- The ignition mechanisms of inert fuel tanks impacted by either inert or energetic fragments are experimental investigated in this study. The results of high-speed video of fragment impact on the fuel tank, mathematical expressions for the ignition mechanisms, and variation curves of gas and oxygen concentrations and flight time with impacting velocity of the fragments are presented, which might provide helpful references for academicians in the relevant fields. However, the manuscript is not well prepared and a few typo and grammatical errors appear. At least minor modification should be carried out prior to its being further consideration for possible publication in the journal.
- Typo or grammatical errors can be easily found in the text. In the title and lines 20 and 42, “inerted” could be “inert” or other equivalent. At line 25, it could be “In addition”; at line 19, it could be “liquid fuel”; at line 17, abbreviations such as “LOC” generally do not appear in the abstract and their full words should be provided once their corresponding abbreviations appear, for example HEI at line 82, and LEL and UEL at line 100; at line 29, it could be “which”; at line 44, it could be “cm3“; at line 261, it could be “No 12”; at line 324, it could be “equation (5)”; etc.
- There are too many mathematical symbols used in this manuscript. A list of nomenclatures is suggested to be added to the manuscript.
- The cited references in the text generally do not include the first names of cited authors, for example, the cited references at lines 43 and 45. And it could be Wang et al. [9] at line 44-45. At line 91, the cited reference “Yang Pei [19]” cannot be referred on page 19. Please carefully check the cited references in the text and References at the end of the manuscript.
- The format of references on pages 18-19 should be consistent, for example ref. [16] and follows the requirements of the journal’s references.
- Please provide the data source for the physicochemical properties of RP-3 and JP-8 at Table 1 on page 4. What is the unit of β at line 364? And what is the reference source of equation (17) at line 372? At the ordinate of Figure 15, what is the meaning of lg and the unit of flight time?
- What are the reactive compounds, respectively in the energetic fragments of No. 10-15 in Table 2 on page 7? How much chemical energy is released from the impact of the energetic fragments onto the fuel tank?
- At line 354, k should be “conductivity coefficient” rather than “kinematic viscosity coefficient”. At line 359, the unit of kinematic viscosity coefficient should be “mm2/s” rather than “kg/(m.s)”.
Author Response
Response to Reviewer 2 Comments
Point 1: The ignition mechanisms of inert fuel tanks impacted by either inert or energetic fragments are experimental investigated in this study. The results of high-speed video of fragment impact on the fuel tank, mathematical expressions for the ignition mechanisms, and variation curves of gas and oxygen concentrations and flight time with impacting velocity of the fragments are presented, which might provide helpful references for academicians in the relevant fields. However, the manuscript is not well prepared and a few typo and grammatical errors appear. At least minor modification should be carried out prior to its being further consideration for possible publication in the journal.
Response 1: It is my honor for your approval, and I will revise my manuscript carefully according to the reviewer's suggestion.
Point 2: Typo or grammatical errors can be easily found in the text. In the title and lines 20 and 42, “inerted” could be “inert” or other equivalent. At line 25, it could be “In addition”; at line 19, it could be “liquid fuel”; at line 17, abbreviations such as “LOC” generally do not appear in the abstract and their full words should be provided once their corresponding abbreviations appear, for example HEI at line 82, and LEL and UEL at line 100; at line 29, it could be “which”; at line 44, it could be “cm3“; at line 261, it could be “No 12”; at line 324, it could be “equation (5)”; etc.
Response 2: We are so sorry for our typo or grammatical errors in the text. And I have revised them carefully as suggested by the reviewer and marked them red in the paper.
Point 3: There are too many mathematical symbols used in this manuscript. A list of nomenclatures is suggested to be added to the manuscript.
Response 3: Nomenclatures is added to the manuscript in the end according to the suggestions of reviewer.
Point 4: The cited references in the text generally do not include the first names of cited authors, for example, the cited references at lines 43 and 45. And it could be Wang et al. [9] at line 44-45. At line 91, the cited reference “Yang Pei [19]” cannot be referred on page 19. Please carefully check the cited references in the text and References at the end of the manuscript.
Response 4: First, we are sorry for our incorrect expressions of citations that first names of cited authors appeared in the text. And we have corrected it. Then Yang Pei in reference 19 is the corresponding author. To avoid confusion, we have corrected it in line 94 carefully checked the cited references in the text.
Point 5: The format of references on pages 18-19 should be consistent, for example ref. [16] and follows the requirements of the journal’s references.
Response 5: We are sorry for the reference issues. And we have changed the reference format to be consistent with the requirements of journal.
Point 6: Please provide the data source for the physicochemical properties of RP-3 and JP-8 at Table 1 on page 4. What is the unit of β at line 364? And what is the reference source of equation (17) at line 372? At the ordinate of Figure 15, what is the meaning of lg and the unit of flight time?
Response 6: We have provided the sources of the physicochemical properties of RP-3 and JP-8 at the Table 1 of the paper according to the Reviewer’s comment. β is the coefficient of energy transfer when internal energy is converted into kinetic energy. So there is no specific unit forβ. Equation (17) is derived from that the amount of substance is equal to the volume divided by the molar volume and mass divided by molar mass as shown.
In this paper, the values of ignition delay time and flight time of fragments are very small. Therefore, the logarithm is taken for all of them. The purpose of taking the logarithm is to compare the magnitudes more conveniently. So the ordinate of Figure 15 has no special practical significance.
Point 7: What are the reactive compounds, respectively in the energetic fragments of No. 10-15 in Table 2 on page 7? How much chemical energy is released from the impact of the energetic fragments onto the fuel tank?
Response 7: The reactive compounds in the energetic fragments of No. 10-15 are mainly aluminum and Poly tetra fluoroethylene (PTFE). Chemical energy released from energetic fragments mainly related to impact speed. Because the degree of reaction of the reactive species is affected by the impact speed. According to the experimental results and ref [5] and ref [13] in this paper, energetic fragments will release about 3 times the chemica energy of its own kinetic energy in this speed range.
Point 8: At line 354, k should be “conductivity coefficient” rather than “kinematic viscosity coefficient”. At line 359, the unit of kinematic viscosity coefficient should be “mm2/s” rather than “kg/(m.s)”.
Response 8: We are sorry for errors. And we have made corrections based on the reviewer's comments. And these corrections are marked in red in the papers.
Special thanks to you for your good comments.

Reviewer 3 Report
The authors examine the impact of inert and energetic fragments to an aviation fuel tank at varying speeds. Using appropriate experimental techniques, the authors draw conclusions and evidence as to the necessary criteria for tank disassembly and ignition. Although fuel tank failure has been well studied and mitigation techniques have been incorporated, this manuscript examines fragment mass and speed as new considerations and provide a potential model to assess impact of targeted threats. This paper is well organized and written and the experimental process and results are of significant interest. This paper should absolutely be considered for publication with only minor revisions. See three comments:
- In Table 2 test 10/11 is the text meant to read “the fuel tank was completely destroyed”?
- Can the authors add a conclusion or discussion about the disassembly mode? From the notes in Table 2, seems like common failure mode was the lid popping off of the fuel tank, insight may lead to improvements in fuel tank design to minimize disassembly.
- 13 and 14 should be explained more in their accuracy of prediction to measured data. This is done for following figures, but not at this stage. Tests 8 and 9 are the only to fall close to the ignitability region, only test 9 ignites, explain potential inconsistencies between model and test 8.
Author Response
Response to Reviewer 3 Comments
Point 1: The authors examine the impact of inert and energetic fragments to an aviation fuel tank at varying speeds. Using appropriate experimental techniques, the authors draw conclusions and evidence as to the necessary criteria for tank disassembly and ignition. Although fuel tank failure has been well studied and mitigation techniques have been incorporated, this manuscript examines fragment mass and speed as new considerations and provide a potential model to assess impact of targeted threats. This paper is well organized and written and the experimental process and results are of significant interest. This paper should absolutely be considered for publication with only minor revisions. See three comments.
Response 1: It is my honor for your approval, and I will revise my manuscript carefully according to the reviewer's suggestion.
Point 2: In Table 2 test 10/11 is the text meant to read “the fuel tank was completely destroyed”?
Response 2: We are sorry for your confusion about my expression. But the fuel tank was relatively intact rather than “completely destroyed”.And we modified our expression in the text.
Point 3: Can the authors add a conclusion or discussion about the disassembly mode? From the notes in Table 2, seems like common failure mode was the lid popping off of the fuel tank, insight may lead to improvements in fuel tank design to minimize disassembly.
Response 3: In this paper, the tested tank was filled with 60% oil and placed on a platform. The placement of the fuel tanks may differ from the aircraft. After the high-speed impact of fragments, there are 3 modes of damage to the fuel tank according to the test results, First, the fuel tank has only perforations on the penetrating surface and the back, and only small cracks occur at the welds. The overall structure is complete, as shown in the test results of No. 1-6. Second, the fuel tank is greatly deformed, the weld is severely cracked, and the upper cover is opened, such as the test results of No. 8 and No. 13-15. The third type, the fuel tank is completely disintegrated, such as the No. 9 test result. The oxygen concentration in the fuel tank will reach the ignition range, and the fuel can be ignited only in the second and third damage modes, which is mainly related to the kinetic energy of the fragments. Therefore, the test results can also help aircraft fuel tank designers lead to improvements to minimize disassembly.
Point 4: 13 and 14 should be explained more in their accuracy of prediction to measured data. This is done for following figures, but not at this stage. Tests 8 and 9 are the only to fall close to the ignitability region, only test 9 ignites, explain potential inconsistencies between model and test 8.
Response 4: The comment is very meaningful. Figure 13 and Figure 14 are the variations of oil gas and oxygen concentration inside the fuel tank when the fuel tank is not disassembled (unshaded parts). And we have added an explanation of the variations to the prediction accuracy of the test results in the paper.
Two conditions must be met at the same time. The flight time of the fragments is greater than the combustion delay time for fuel and within the oxygen concentration range. In the No. 8 and No. 9 tests, the fuel tank was disintegrated, and this variation is not suitable to the test results. The oxygen could support the fuel ignition in tests Nos. 8 and 9. However, the flight time of the fragment was less than the ignition delay time of the oil gas for the No. 8 test. So in Test No. 8, the fuel can not be ignited. But it was the opposite in test No.9. The model and experimental results are explained in detail in lines 438 to 443 in the paper.
Special thanks to you for your good comments.

Round 2
Reviewer 1 Report
Only in this form is acceptable.